# A Systematic Review of High-Dose Methotrexate for Adults with Primary Central Nervous System Lymphoma

**DOI:** 10.3390/cancers15051459

**Published:** 2023-02-25

**Authors:** Gabriela Villanueva, Martin Guscott, Paula Schaiquevich, Claudia Sampor, Ryan Combs, Nicolás Tentoni, Miriam Hwang, Jennifer Lowe, Scott Howard

**Affiliations:** 1Resonance, Inc., 5239 Jeffrey Keith Drive, Arlington, TN 38002, USA; 2Laboratory of Applied Statistics in the Health Sciences, Faculty of Medicine, University of Buenos Aires, Paraguay 2155, Buenos Aires C1121 ABG, Argentina

**Keywords:** primary central nervous system lymphoma, high-dose methotrexate, rituximab, overall response rate, progression-free survival

## Abstract

**Simple Summary:**

High-dose methotrexate (HDMTX) is the backbone of induction therapy for primary central nervous system lymphoma (PCNSL). There are numerous different protocols to treat PCNSL that incorporate a wide range of HDMTX doses and various combinations with other chemotherapeutic agents. This systematic review was conducted to summarize the various treatment regimens for PCNSL and determine outcomes among the different doses of HDMTX and combination regimens. The findings are intended to provide guidance on the optimal dose and regimen of HDMTX for the treatment of PCNSL.

**Abstract:**

Primary central nervous system lymphoma (PCNSL) is a highly aggressive non-Hodgkin lymphoma that is confined within the CNS. Due to its ability to cross the blood–brain barrier, high-dose methotrexate (HDMTX) is the backbone for induction chemotherapy. This systematic review was conducted to observe outcomes among different HDMTX doses (low, <3 g/m^2^; intermediate, 3–4.9 g/m^2^; high, ≥5 g/m^2^) and regimens used in the treatment of PCNSL. A PubMed search resulted in 26 articles reporting clinical trials using HDMTX for PCNSL, from which 35 treatment cohorts were identified for analysis. The median dose of HDMTX used for induction was 3.5 g/m^2^ (interquartile range IQR, 3–3.5); the intermediate dose was most frequently used in the studies examined (24 cohorts, 69%). Five cohorts used HDMTX monotherapy, 19 cohorts used HDMTX + polychemotherapy, and 11 cohorts used HDMTX + rituximab ± polychemotherapy. Pooled overall response rate (ORR) estimates for low, intermediate, and high dose HDMTX cohorts were 71%, 76%, and 76%, respectively. Pooled 2-year progression-free survival (PFS) estimates for low, intermediate, and high HDMTX dose cohorts were 50%, 51%, and 55%, respectively. Regimens that included rituximab showed a tendency to have higher ORR and 2-year PFS than those that did not include rituximab. These findings indicate that current protocols utilizing 3–4 g/m^2^ of HDMTX in combination with rituximab provide therapeutic efficacy in PCNSL.

## 1. Introduction

Primary central nervous system lymphoma (PCNSL) is a highly aggressive non-Hodgkin lymphoma that is confined to the central nervous system (CNS) and vitreoretinal space. It is a rare malignancy that accounts for 4% of intracranial neoplasms and 4–6% of extra-nodal lymphomas, and can occur in both immunocompromised and immunocompetent individuals [1,2]. More than 90% of PCNSLs are of diffuse large B-cell lymphoma (DLBCL) phenotype, and ‘primary DLBCL of the CNS’ was recognized as a distinct entity by the 2017 World Health Organization (WHO) classification of hematopoietic and lymphoid tumors [3,4]. PCNSL predominantly affects adults older than 60 years of age (median age of 67 at diagnosis); while the age-standardized incidence is 0.4–0.5 per 100,000/year, those aged 70–79 years have the highest incidence of 4.3 per 100,000/year [5,6,7]. Of concern is the fact that while median overall survival (OS) has doubled over the past 40 years from 12.5 to 25 months in younger patients, median survival in those over 70 years old remains at 6–7 months [7]. Additionally, while tumor regression is achieved in 85% of cases with chemo- and radiation therapy, approximately 15% are refractory and relapse rates are high, with relapses usually occurring within 2 years, but have been reported up to 10–13 years after initial diagnosis [8,9,10]. 

Currently intravenous administration of high-dose methotrexate (HDMTX) is the backbone of PCNSL treatment due to its ability to penetrate the blood–brain barrier (BBB). Methotrexate doses ranging from 1–8 g/m^2^ are used in most regimens, and various combinations with other agents have been studied in clinical trials, resulting in overall response rates (ORR) of 35–74% and a median OS of 25–50 months [10]. Rituximab, a CD20 monoclonal antibody and standard component of non-Hodgkin B-cell lymphomas, has been investigated in clinical trials and retrospective studies for PCNSL since 2010 due its efficacy in treating systemic DLBCL and its minimal adverse effects [11]. Most of the current methotrexate-based polychemotherapy regimens for PCNSL incorporate rituximab: rituximab-HDMTX-vincristine-procarbazine (R-MVP) [12], rituximab-HDMTX-temozolomide (R-MT) [13], rituximab-HDMTX-carmustine-teniposide-prednisolone (R-MBVP) [14], and rituximab-HDMTX-thiotepa-cytarabine (MATRix) [15]. Despite the inclusion of rituximab in various regimens, its efficacy in PCNSL is a subject of ongoing debate due to its large size (145 kD) precluding passage through the BBB and cerebrospinal fluid concentrations measuring only 0.1% of serum concentration following intravenous administration [16,17]. There is, however, some radiographic evidence suggesting that the BBB is disrupted at the site of PCNSL infiltration allowing for some penetration [18].

Due to its rarity, high-quality evidence from clinical trials regarding treatment for PCNSL is scarce and there is a lack of consensus regarding optimal treatment among the various regimens [11]. Furthermore, to date there have been only two randomized studies (one phase 2 and one phase 3) regarding the efficacy of rituximab in PCNSL that reported conflicting results [14,15]. This systematic review was conducted to summarize the treatment protocols for PCNSL and to observe outcomes among different doses of HDMTX and different treatment regimens that include HDMTX in the treatment of PCNSL. 

## 2. Methods

The selection and systematic review of appropriate studies was performed in accordance to the Preferred Reporting Items for Systematic Review and Meta-Analyses (PRISMA) 2020 statement guidelines (Figure 1) [19]. This systematic review was part of a broader review investigating the optimal use of HDMTX in CNS tumors. A Medline/PubMed search was conducted for papers that were published up to July 2021 using Medical Subject Heading (MeSH) search terms determined from the 2021 World Health Organization (WHO) Classification of Tumors of the Central Nervous System [20]. Criteria for inclusion were clinical studies on CNS tumors in humans and use of HDMTX (defined as dose ≥ 500 mg/m^2^). Criteria for exclusion were nonclinical or animal studies and review articles. The initial search identified 587 articles pertaining to the use of HDMTX in CNS malignancies, all of which were written in English, Spanish, or German. Following removal of duplications, four authors (MG, GV, CS, PS) independently assessed article eligibility based on review of abstracts and came to a consensus on the selection of 264 articles. To focus solely on PCNSL, inclusion criteria were refined to include only prospective clinical trials for PCNSL, and retrospective studies, case series, long-term follow-up studies, and articles reporting results on less than 25 patients were excluded. Ultimately, 26 full-text articles of prospective clinical trials reporting the use of HDMTX in PCNSL were reviewed for this analysis. 

### 2.1. Data Collection

Each arm of every randomized trial was identified as a separate cohort of uniformly treated patients, resulting in 35 analytic cohorts extracted from the 26 prospective clinical trials. Information was collected from each cohort to include study type, number of patients, tumor histology (e.g., DLBCL), level of evidence, methotrexate dose, number of cycles and courses, use of rituximab, use of other chemotherapeutic agents for induction, chemotherapy to the CNS compartment (i.e., via intrathecal or intracerebroventricular administration) during induction, and type of consolidation therapy. Additional data were collected on treatment-related toxicities. Outcomes of interest were 2-year progression free survival (PFS) and ORR, where ORR included patients with complete response (CR) or partial response (PR) at the end of induction. In cohorts for which PFS was not specifically reported, PFS data were estimated from the survival curves that were presented or supplementary data. 

### 2.2. Statistical Analysis

HDMTX doses were categorized as low (<3 g/m^2^), intermediate (3–4.9 g/m^2^), and high (≥5 g/m^2^). Induction protocol regimens were grouped as HDMTX monotherapy (Group 1), HDMTX + polychemotherapy (Group 2), and HDMTX + rituximab ± polychemotherapy (Group 3). The number of planned courses administered for induction with HDMTX were categorized as <5 and ≥5, and administration of chemotherapy into the CNS compartment via either intrathecal (IT) or intracerebroventricular (ICV) routes during induction were recorded and classified as binary variables. 

Proportional meta-analyses were applied to estimate the pooled effect of different HDMTX protocols on ORR and 2-year PFS. As heterogeneity among the included studies was expected due to methodological differences, a random effects model was applied; heterogeneity was assessed using a chi-squared heterogeneity test and I-squared statistic. Forest plots were used to display pooled estimates and individual study results in selected protocols. To assess the impact of rituximab on the efficacy of induction therapy for PCNSL, exploratory comparisons of ORR and 2-year PFS estimates by HDMTX dosage categories between cohorts that received rituximab (Group 3) and those that did not receive rituximab (Groups 1 and 2) were conducted.

## 3. Results

From the 26 articles of prospective clinical trials 35 cohorts were identified; a total of 2115 patients comprising the 35 cohorts had received induction with HDMTX and were included in the analysis. Table 1 summarizes the clinical trials and characteristics of each analytical cohort as identified by unique cohort numbers.

### 3.1. Characteristics of the Analytic Cohorts

All 35 cohorts were comprised of adult patients with PCNSL; 30 cohorts included patients older than 65 years of age. Histologically, DLBCL was the most common phenotype accounting for 57% of the study population while other subtypes (e.g., Burkitt lymphoma, T-cell lymphoma) were reported in 2%; there was no histological subtype reported in 41%. Intra-ocular tumors were reported in 55 patients, and 139 patients had leptomeningeal dissemination at diagnosis.

### 3.2. High-Dose Methotrexate Induction Regimens

The median dose of HDMTX used for induction therapy was 3.5 g/m^2^ (IQR, 3–3.5); low dose HDMTX (<3 g/m^2^) was used in 7 cohorts (20%), intermediate dose (3–4.9 g/m^2^) in 24 cohorts (68.6%), and high (≥5 g/m^2^) was used in 4 cohorts (11.4%). The median number of cycles provided was 4 (IQR, 3–5) and the median number of planned courses was 4 (IQR, 4–6); 20 cohorts (57%) reported number of planned courses as <5. Five cohorts used HDMTX monotherapy (Group 1), 19 cohorts used HDMTX + polychemotherapy (Group 2), and 11 cohorts used HDMTX + rituximab ± polychemotherapy (Group 3); 1 cohort in Group 3 used HDMTX + rituximab only. Chemotherapy to the CNS compartment was administered in 14 cohorts (40%).

### 3.3. Overall Response Rates

Pooled ORR estimates for the three HDMTX dosage categories were 71% [95% CI, 52–87%], 76% [95% CI, 68–84%], and 76% [95% CI, 57–91%] for the low (<3 g/m^2^), intermediate (3–4.9 g/m^2^), and high (≥5 g/m^2^) dose cohorts, respectively (Figure 2). Pooled ORR estimates for the three HDMTX induction regimens were as follows: Group 1 (HDMTX monotherapy), 69% [95% CI, 41–91%]; Group 2 (HDMTX + polychemotherapy), 70% [95% CI, 61–79%]; Group 3 (HDMTX + rituximab ± polychemotherapy), 85% [95% CI, 79–90%], showing a trend for increasing ORR with the addition of rituximab (Figure 3). 

ORR estimates for cohorts that received <5 courses and ≥5 courses of HDMTX were 73% and 79%, respectively. Cohorts that included CNS chemotherapy as part of induction therapy had a pooled estimate ORR of 79% compared to 73% in those that did not administer CNS chemotherapy (Table 2). 

**Figure 2 cancers-15-01459-f002:**
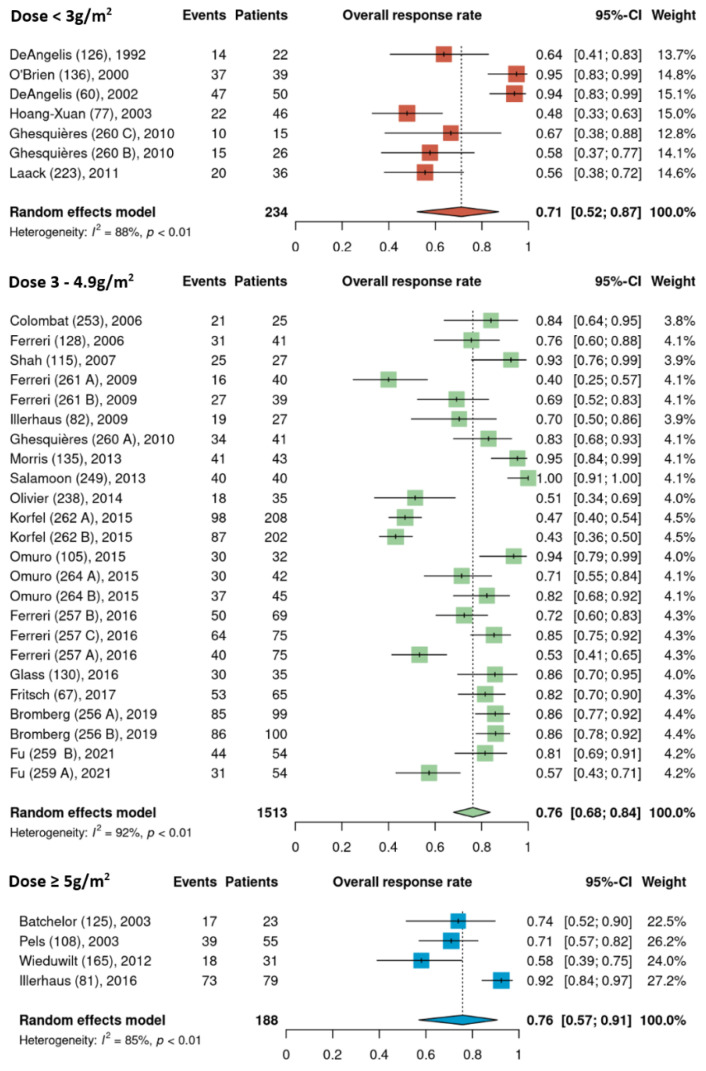
Overall response rates of the different HDMTX dose categories [12,14,15,18,21,22,23,24,25,26,27,28,29,30,31,32,33,34,35,36,37,38,39,40,41,42].

**Figure 3 cancers-15-01459-f003:**
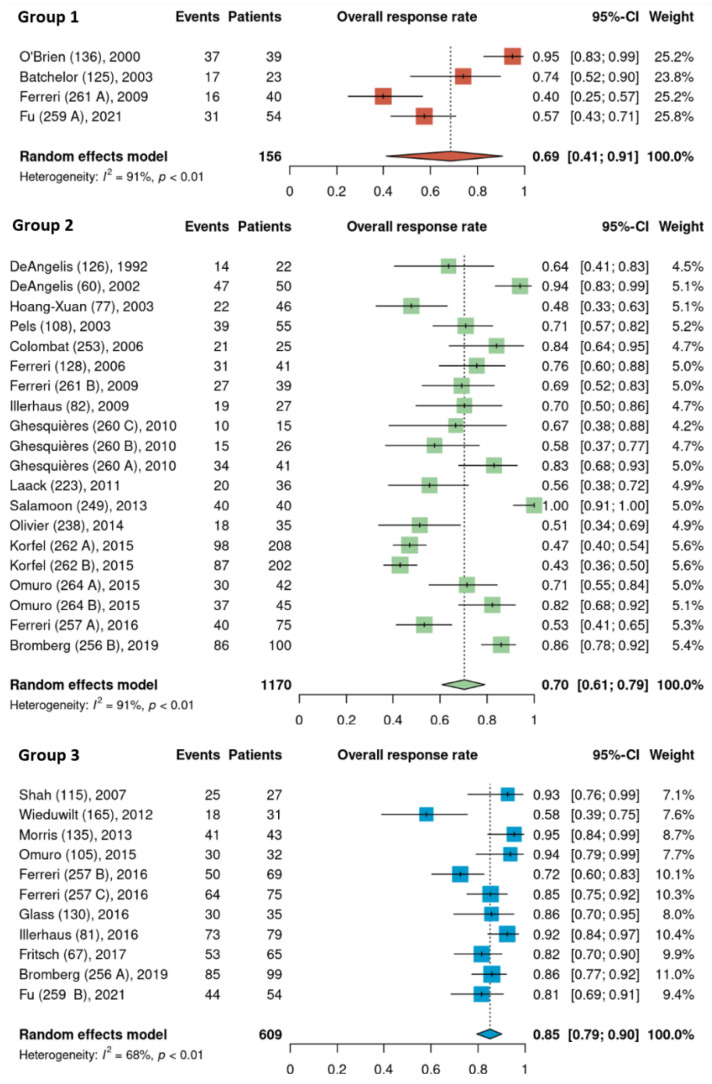
Overall response rates of the different HDMTX induction regimens. Group 1, HDMTX monotherapy. Group 2, HDMTX + polychemotherapy. Group 3, HDMTX + rituximab ± polychemotherapy [12,14,15,18,21,22,23,24,25,26,27,28,29,30,31,32,33,34,35,36,37,38,39,40,41,42].

Pooled ORR estimates of cohorts that received rituximab and those that did not were analyzed separately among the different HDMTX dosage categories. None of the low dose HDMTX cohorts received rituximab. For those receiving intermediate dose HDMTX (3–4.9 g/m^2^), the ORR of 86% [95% CI, 81–90%] in cohorts that received rituximab was higher than in cohorts that did not receive rituximab, at 69% [95% CI, 56–79%]. Among the high dose HDMTX cohorts, the ORR was 78% [95% CI, 38–100%] in those that received rituximab and 72% [95% CI, 61–82%] for those that did not receive rituximab (Figure 4). 

### 3.4. Progression-Free Survival

Pooled 2-year PFS estimates for the three HDMTX dosage categories were 50% [95% CI, 38–62%], 51% [95% CI, 44–58%], and 55% [95% CI, 31–78%] for the low, intermediate, and high dose cohorts, respectively (Figure 5). Pooled 2-year PFS estimates for the three HDMTX induction regimens were as follows: Group 1, 43% [95% CI, 26–60%]; Group 2, 48% [95% CI, 39–56%]; Group 3, 59% [95% CI, 50–66%), showing a trend for increasing 2-year PFS with the addition of rituximab (Figure 6). 

The pooled 2-year PFS estimates for cohorts that received <5 courses and ≥5 courses of HDMTX were 50% and 52%, respectively. Cohorts that included CNS chemotherapy as part of induction therapy had a pooled 2-year PFS estimate of 52%, while it was 51% for those that did not receive CNS chemotherapy. (Table 2). 

**Figure 5 cancers-15-01459-f005:**
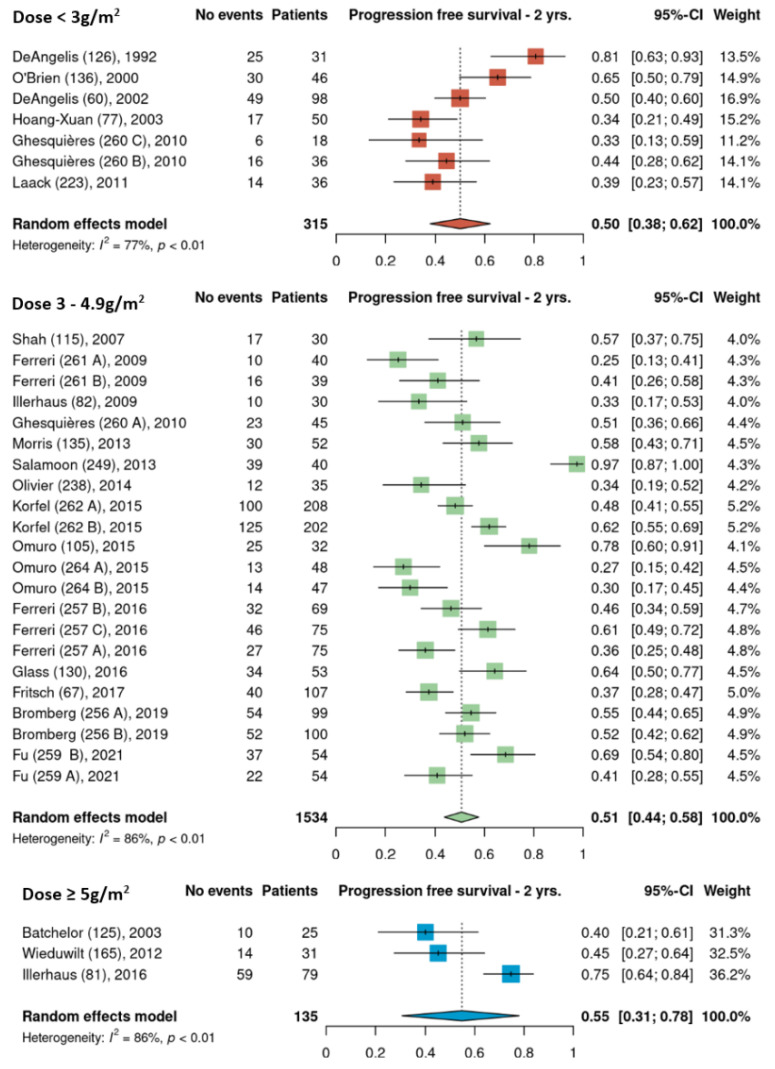
The 2-year progression-free survival of the different HDMTX dose categories [12,14,15,18,21,23,24,26,27,28,29,30,31,32,33,34,35,36,37,38,39,41,42].

**Figure 6 cancers-15-01459-f006:**
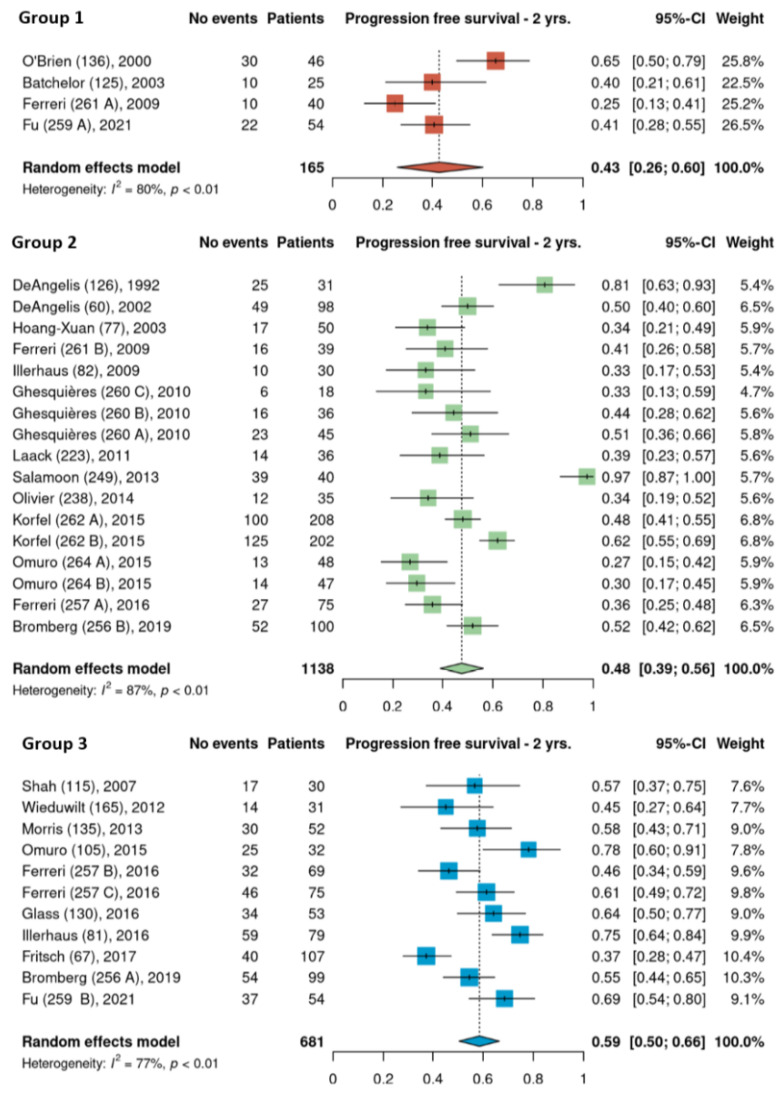
The 2-year progression-free survival of the different HDMTX induction regimens. Group 1, HDMTX monotherapy. Group 2, HDMTX + polychemotherapy. Group 3, HDMTX + rituximab ± polychemotherapy [12,14,15,18,21,23,24,26,27,28,29,30,31,32,33,34,35,36,37,38,39,41,42].

Pooled estimates of 2-year PFS of cohorts that received rituximab and those that did not were analyzed separately among the different HDMTX dosage categories. None of the low dose HDMTX cohorts received rituximab. For those receiving intermediate dose HDMTX (3–4.9 g/m^2^), 2-year PFS was 58% [95% CI, 49–66%] in cohorts that received rituximab and 45% [95% CI, 35–66%] in cohorts that did not received rituximab. Among the high dose HDMTX cohorts, 2-year PFS was 61% [95% CI, 32–87%] in those that received rituximab and 40% [95% CI, 21–61%] for those that did not receive rituximab (Figure 7). Although there were four total cohorts that used high dose HDMTX, i.e., two in the rituximab (+) group and two in the rituximab (−) groups, 2-year PFS data was missing from one cohort from the rituximab (−) group.

### 3.5. Consolidation Therapy

Consolidation therapy was provided in 29 of the 35 cohorts. Radiation therapy (RT) only was given in 11 cohorts, high-dose chemotherapy (HDCT) only was given in 4 cohorts, stem cell transplantation (SCT) only was used in 2 cohorts, RT + HDCT in 6 cohorts, RT + SCT in 1 cohort, and SCT or RT in 3 cohorts. Cytarabine was used for HDCT consolidation in most cohorts; HDMTX was given in cohort number 125 [21], temozolamide was given in cohort number 130 [30], and cytarabine + etoposide was administered in cohort number 165 [42] (Table 1). RT dosage ranged from 25 to 54 Gy; pooled estimates for ORR and 2-year PFS appear similar in cohorts that did and did not receive RT (Table 2). In the 6 cohorts that received SCT (alone or in combination), ORR and 2-year PFS showed a tendency to be lower than the cohorts that did not receive SCT (Table 2).

### 3.6. Toxicities 

Treatment-induced toxicities were reported in 32 of the 35 cohorts while the remaining 3 cohorts provided no information regarding toxicities. Renal toxicity was most common with 30 cohorts reporting its occurrence, among which 22 cohorts reported grade 3–4 toxicity and 5 reported grade 1–2; severity was not specified in 3 cohorts. Neurotoxicity was reported in 18 cohorts, among which 10 cohorts reported grade 3–4 neurotoxicity. Although the exact timing of its occurrence (i.e., after induction with HDMTX or after consolidation therapy) was not always specified, 9 cohorts reporting neurotoxicity included IT or ICV chemotherapy and 14 cohorts reported RT as a component of consolidation. Mucositis was reported in 15 cohorts, all of which were of grade 3–4 severity. 

## 4. Discussion

This systematic review aimed to summarize the treatment protocols and outcomes of clinical trials for PCNSL with respect to dose of HDMTX and different regimens of HDMTX (i.e., HDMTX monotherapy, HDMTX + polychemotherapy, and HDMTX + rituximab ± polychemotherapy). Twenty-six articles reporting on clinical trials using HDMTX were reviewed, from which 35 treatment cohorts were identified and used for analysis. 

The dose of HDMTX used in protocols ranged widely from 1 to 8 g/m^2^, with a median dose of 3 g/m^2^. Based on this median value and the doses used across the 35 cohorts (Table 1), HDMTX dose categories were defined as low dose (<3 g/m^2^), intermediate dose (3 to 4.9 g/m^2^), and high dose (≥5 g/m^2^) to conduct analysis. (As HDMTX was administered intravenously in all cohorts, peak plasma methotrexate levels were proportional to the HDMTX dose [43]. This suggests that outcomes are closely related to the dose of methotrexate.) The intermediate dose was most frequently used as it was reported in 24 of the 35 cohorts (69%); however, pooled ORR estimates were 71%, 76%, and 76% for the low, intermediate, and high dose cohorts, respectively, showing no difference among HDMTX dose categories (Figure 2). Similarly, there was no difference in the pooled estimates for 2-year PFS among the dose categories: 50% (low dose), 51% (intermediate dose), and 55% (high dose) (Figure 3). These findings suggest that higher doses of HDMTX may not be necessary and the current mainstay of 3 to 3.5 g/m^2^ is sufficient in achieving treatment efficacy while reducing the risk of severe toxicities that may arise from use of higher doses, especially in individuals with reduced renal function [44,45]. The publication dates for the articles included in this review span a period between 1992 (DeAngelis et al.) [23] and 2021 (Fu et al.) [28]. Over the nearly 30 years that encompass this period, there was no significant linear trend found between HDMTX dose and the chronological passage of time; most regimens used 3–3.5 g/m^2^ reflecting the fact that this dose has been and continues to be efficacious and safe in the treatment of PCNSL. The relatively smaller number of cohorts that used <3 g/m^2^ (n = 7) or ≥ 5g/m^2^ (n = 4) of HDMTX in this review, however, limited the ability to fully assess the impact that lower or higher doses of HDMTX might have on treatment efficacy. In a study determining the optimal dose of HDMTX in 50 patients with PCNSL, Dalia et al. reported no significant difference in PFS or OS when comparing HDMTX doses of 8 g/m^2^ to 3.5 g/m^2^, which was concordant with our results [46]. They also reported that neither HDMTX dose reductions or higher cumulative HDMTX doses were associated with significant differences in PFS or OS, which further support our results. Contrary to these findings, in a comparison of outcomes using 3.5 g/m^2^ (n = 32) and 8 g/m^2^ (n = 41) in patients with PCNSL, Li et al. reported significantly greater rates of CR (68.3% vs 43.8%, *p* = 0.03) and longer median PFS (17 months vs 9 months, *p* = 0.03) in patients receiving 8 g/m^2^ compared to those receiving 3.5 g/m^2^ [47]. These results, however, should be interpreted with caution as the median age and IQR were significantly lower in the 8 g/m^2^ group at 49 years [IQR 42–55] compared to the 3.5 g/m^2^ group at 61 years [IQR 51–69] (*p* = 0.01), as age is a significant prognostic factor for PCNSL [48]. Li et al. did further compare outcomes between the two dose categories only in patients younger than 65 years of age and found a higher median PFS of 17.7 months in the 8 g/m^2^ group compared to 7 months in the 3.5 g/m^2^ group (*p* = 0.02). 

With respect to induction regimens, 31 cohorts (89%) used at least one other chemotherapeutic agent in combination with HDMTX and only 4 cohorts utilized HDMTX monotherapy, reflecting the prevalent use of polychemotherapy in treatment for PCNSL [47]. Alkylating agents (cyclophosphamide, lomustine, thiotepa, etc.) and the antimetabolite cytarabine were most frequently incorporated as they were used in 21 cohorts and 22 cohorts, respectively. Other agents used were vinca alkaloids (10 cohorts), corticosteroids (11 cohorts), topoisomerase inhibitors (5 cohorts), and anthracyclines (5 cohorts) (Table 1). Rituximab was utilized with HDMTX induction in 11 of the 35 cohorts (31%), among which one cohort (259B) [28] used only rituximab in combination with HDMTX. Pooled estimates for ORR were 69% for cohorts that utilized HDMTX monotherapy, 72% for HDMTX + polychemotherapy, and 84% for HDMTX + rituximab ± polychemotherapy, showing a trend for increasing ORR with the addition of rituximab. Pooled 2-year PFS estimates for HDMTX monotherapy, HDMTX + polychemotherapy, and HDMTX + rituximab ± polychemotherapy were 43%, 48%, and 59%, respectively, which also revealed a trend for increasing 2-year PFS when rituximab was added to the induction protocol. 

To further examine the impact of rituximab, exploratory comparisons of ORR and 2-year PFS were conducted between cohorts that received rituximab (regimen Group 3) and those that did not receive rituximab (regimen Groups 1 and 2) among the different HDMTX dose categories. Among cohorts using the intermediate dose (3–4.9 g/m^2^), pooled ORR estimates were 86% in cohorts that received rituximab which was higher than the 69% found in cohorts that did not receive rituximab (Figure 4). Among cohorts using high dose HDMTX (≥ 5 g/m^2^), pooled ORR estimates were 73% for those receiving rituximab and 72% for those that did not receive rituximab. The small number of cohorts that used high dose HDMTX in both groups (two cohorts each), however, likely did not provide adequate power to fully represent any potential effect the higher dose might have had on ORR. A similar pattern was found when comparing pooled 2-year PFS estimates among the cohorts that used the intermediate dose (3–4.9 g/m^2^): 58% in cohorts that received rituximab and 45% in those that did not receive rituximab (Figure 7). Although statistical comparisons could not be performed due to the lack of patient level data, the higher ORR and 2-year PFS in cohorts that received rituximab imply that the addition of rituximab to HDMTX-based induction regimen may positively impact treatment outcomes of PCNSL. To date there have been two prospective randomized studies investigating the efficacy of rituximab in PCNSL: the IELSG32 trial [15] comparing three arms (A, HDMTX + cytarabine; B, HDMTX + cytarabine + rituximab; C, HDMTX + cytarabine + rituximab + thiotepa) and the HOVON105/ALLG NHL24 trial [14] which compared MBVP with and without rituximab. These two trials, however, resulted in conflicting findings, where addition of rituximab led to improved outcomes in IELSG 32, but no significant difference in HOVON105/ALLG NHL24. Using these two trials, Schmidt et al. conducted a trial-level random-effects meta-analysis to determine whether addition of rituximab would impact OS and PFS. They found that OS was not significantly improved as determined by a hazard rate (HR) of death in the pooled analysis at 0.76 [95% CI, 0.52–1.12], but reported rituximab may improve PFS with HR for PFS at 0.65 [95% CI, 0.45–0.95], albeit with low certainty of evidence [49]. Similarly, Fritsch et al. reported a single center prospective phase II study in 28 elderly patients (age ≥ 65) and found that the addition of rituximab to HDMTX + lomustine + procarbazine (R-MCP) improved PFS but not OS compared to MCP [50].

### Limitations

A limitation of this review is the lack of patient-level data which precluded not only statistical comparison among the cohort categories but also multivariate analyses that would have elucidated predictors of outcomes and confounding variables. This lack of patient-level data also limited our reporting of treatment-related toxicity results to the number of cohorts (and not the actual percentage of patients) that reported their occurrences and did not allow for assessment of their impact on outcomes. Similarly, we reported only planned courses of HDMTX induction therapy rather than the actual number of courses because patient-level data were not available. Further, age and performance status are important predictors of prognosis as well as determinants of consolidation therapy modality; the absence of these data in addition to the heterogeneity of modalities made it difficult to assess the impact of consolidation therapy on outcomes. Thus, we were able to only present a descriptive comparison of pooled estimates of ORR and 2-year PFS comparison by number of planned HDMTX courses (<5 vs ≥5), CNS chemotherapy at induction, and provision of RT and SCT without statistical analysis (Table 2). Additionally, there was considerable variability among the treatment cohorts with respect to the administration of leucovorin rescue (specific dose, timing, duration) which would have potentially impacted the frequency of toxicities as well as treatment efficacy (Appendix A).

The outcomes of interest in this review were ORR (CR + PR) and 2-year PFS. Although OS would have provided a more comprehensive assessment of outcomes for the treatment regimens, ORR and 2-year PFS were selected as the main outcomes in this analysis because they were the most frequently reported in the studies included in this review. ORR and 2-year PFS allowed for determination of the immediate effect of HDMTX induction regimens and overall therapeutic efficacy during the first 2 years after diagnosis. 

Another limitation is the relative paucity of articles that reported clinical trials in elderly patients, particularly those over the age of 65. Most of the trials in this review were composed of patients with a wide age range varying from 18 to 85, with the median age for the trials ranging from 41 to 63 years; two trials (cohort numbers 253 and 257) had no patients over the 65 years of age (Appendix A). Only three trials (cohort numbers 67, 77, 82) had patients with a median age ≥ 70 years, and it is notable that no consolidation therapy was given in these trials, which in turn may have impacted PFS [27,31,32]. Although this limitation is likely due to the inherently smaller number of clinical trials of PCNSL that include only elderly patients, it is possible that our results do not properly represent outcomes in this important older age group who are known to have poorer prognoses than younger patients [7,51].

## 5. Conclusions

This systematic review summarized prospective clinical trials utilizing HDMTX for the treatment of PCNSL and assessed outcomes with respect to HDMTX dose and combination regimens used for induction therapy. ORR and 2-year PFS were similar for all three HDMTX dose categories (low, <3 g/m^2^; intermediate 3–4.9 g/m^2^; high, ≥5 g/m^2^), and the intermediate dose, specifically 3–4 g/m^2^ was most commonly used. HDMTX regimens that included rituximab showed a tendency to have higher ORR and 2-year PFS compared to those that did not include rituximab. These findings add to the preliminary evidence supporting that sufficient doses and cycles of HDMTX with the inclusion of rituximab provide therapeutic efficacy for the treatment of PCNSL. Increased efforts are needed to include elderly patients ≥70 years of age in clinical trials to assess therapeutic safety and efficacy of different HDMTX doses, combination chemotherapy, and consolidation modalities.

## Figures and Tables

**Figure 1 cancers-15-01459-f001:**
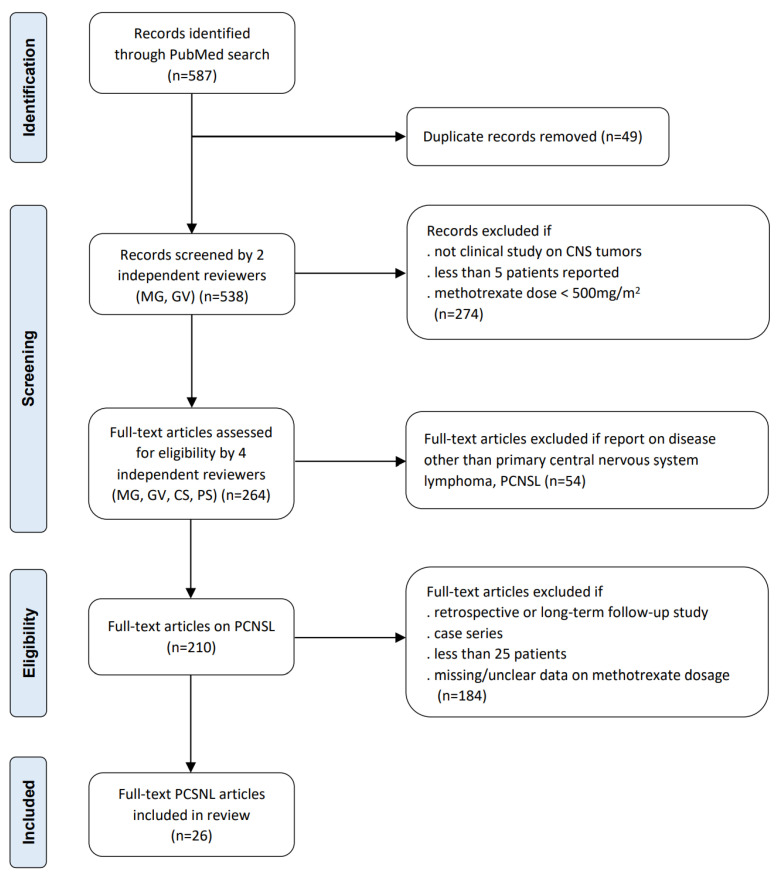
Flow diagram of systematic review according to PRISMA guidelines.

**Figure 4 cancers-15-01459-f004:**
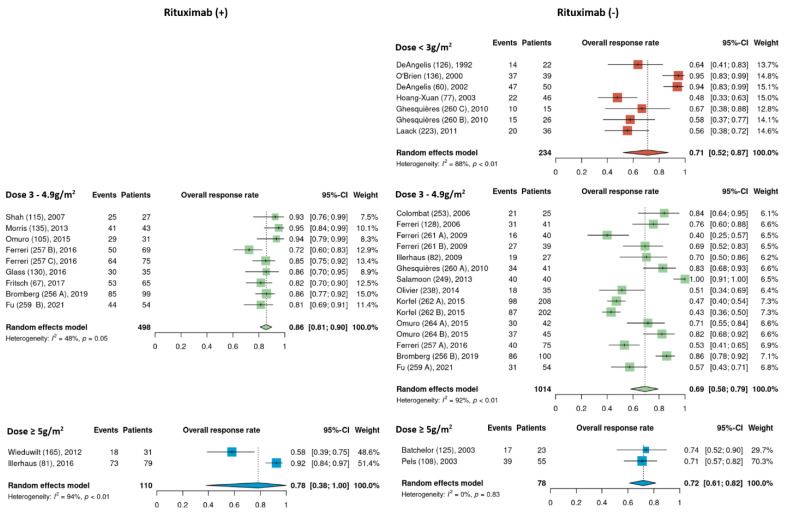
Exploratory comparison of overall response rates between cohorts that did and did not receive rituximab among the different HDMTX dose categories [12,14,15,18,21,22,23,24,25,26,27,28,29,30,31,32,33,34,35,36,37,38,39,40,41,42].

**Figure 7 cancers-15-01459-f007:**
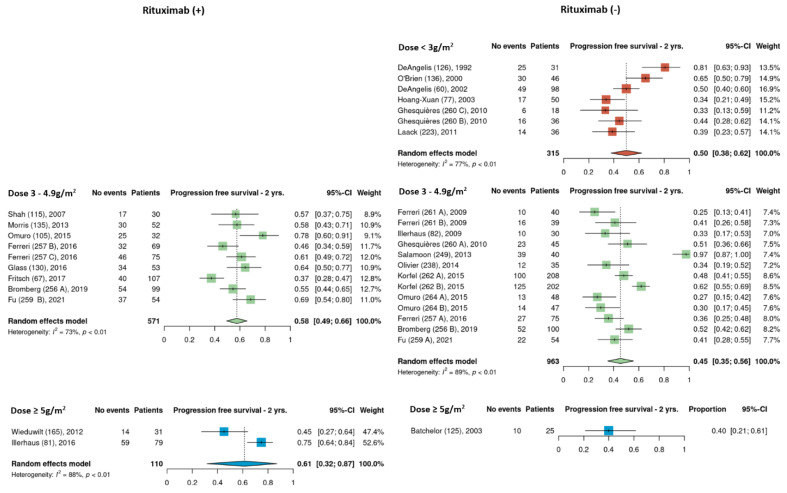
Exploratory comparison of 2-year progression-free survival between cohorts that did and did not receive rituximab among the different HDMTX dose categories [12,14,15,18,21,23,24,26,27,28,29,30,31,32,33,34,35,36,37,38,39,41,42].

**Table 1 cancers-15-01459-t001:** Summary of the treatment protocols and characteristics of the analytic cohorts.

Author (Year)	Cohort Number ^†^	Study Type	Total N^0^ Treated with HDMTX	HDMTX Dose (g/m^2^)	N^0^ of HDMTX Cycles	Rituximab (Y/N)	Other Chemotherapeutic Agents	CNS Delivery	Consolidation Therapy	ORR (CR + PR)	1-Year PFS	2-Year PFS	3-Year PFS
Batchelor et al., (2003) [21]	125	Multicenter, Phase II	25	8	Up to 8	N	-	-	HDCT (MTX)	74%	50%	38%	38%
Bromberg et al., (2019) [14]	256A	Randomized Phase III	99	3	2	Y	Carmustine, teniposide, prednisone	IT ^a^	HDCT (ARA-C) ± RT	86%	65%	55%	55%
Bromberg et al., (2019) [14]	256B	Randomized Phase III	100	3	2	N	Carmustine, teniposide, prednisone	IT ^a^	HDCT (ARA-C) ± RT	86%	58%	52%	40%
Columbat et al., (2006) [22]	253	Multicenter, Phase II	25	3	2	N	Etoposide, carmustine, MPD	IT ^b^	SCT	84%	-	-	-
DeAngelis et al., (1992) [23]	126	Single center	31	1	1	N	Dexamethasone	ICV ^a^	RT + HDCT (ARA-C)	64%	90% ^#^	80% ^#^	66% ^#^
DeAngelis et al., (2002) [24]	60	Single-arm	98	2.5	5	N	Vincristine, procarbazine	ICV ^a^	RT + HDCT (ARA-C)	94%	64%	50%	41%
Ferrari et al., (2016) [15]	257A	Randomized Phase II	75	3.5	4	N	Cytarabine	-	SCT or RT *	54%	40%	36%	31%
Ferrari et al., (2016) [15]	257B	Randomized Phase II	69	3.5	4	Y	Cytarabine	-	SCT or RT *	73%	57%	46%	40%
Ferrari et al., (2016) [15]	257C	Randomized Phase II	75	3.5	4	Y	Cytarabine, thiotepa	-	SCT or RT *	86%	65%	61%	52%
Ferrari et al., (2006) [25]	128	Phase II	41	3.5	3	N	Idarubicin, cytarabine, thiotepa	-	RT	76%	-	-	43%
Ferrari et al., (2009) [26]	261A	Randomized Phase II	40	3.5	4	N	-	-	RT	41%	30% ^¶^	25% ^¶^	21% ^¶^
Ferrari et al., (2009) [26]	261B	Randomized Phase II	39	3.5	4	N	Cytarabine	-	RT	69%	50% ^¶^	40% ^¶^	38% ^¶^
Fritsch et al., (2017) [27]	67	Multicenter Single-arm	107	3	3	Y	Procarbazine, lomustine	-	-	81% ^ǁ^	46%	37%	37%
Fu et al., (2021) [28]	259A	Randomized	54	3	4	N	-	-	RT	57%	46%	40%	15%
Fu et al.,(2021) [28]	259B	Randomized	54	3	4	Y	-	-	RT	81%	70%	68%	28%
Ghesquières et al., (2010) [29]	260A	Multicenter Phase II	45	3	4	N	Cyclophosphamide, doxorubicin, vincristine, prednisone, cytarabine	IT ^c^	RT	82%	68%	51%	45%
Ghesquières et al., (2010) [29]	260B	Multicenter Phase II	36	1.5	4	N	Cyclophosphamide, doxorubicin, vincristine, prednisone, cytarabine	IT ^c^	RT	58%	54%	45%	38%
Ghesquières et al., (2010) [29]	260C	Multicenter Phase II	18	1.5	4	N	Cyclophosphamide, etoposide	IT ^c^	RT	66%	50%	35%	18%
Glass et al.,(2016) [30]	130	Phase I and II	53	3.5	5	Y	Temozolamide	-	RT + HDCT (TMZ)	85%	80%	64%	57%
Hoang-Xuan et al., (2003) [31]	77	Multicenter Phase II	50	1	1	N	Procarbazine, lomustine, MPD	IT^d^	-	48%	40%	34%	30%
Illerhaus et al., (2009) [32]	82	Pilot & Phase II	30	3	3	N	Procarbazine, lomustine	-	-	70%	45%	35%	32%
Illerhaus et al., (2016) [33]	81	Single-arm Phase II	79	8	1	Y	Cytarabine, thiotepa	-	SCT	92%	79%	75%	67%
Korfel et al., (2015) [34]	262A	Randomized Phase III	208	4	6	N	Ifosfamide	-	-	47% ^‡^	72% ^‡^	48% ^‡^	39% ^‡^
Korfel et al., (2015) [34]	262B	Randomized Phase III	202	4	6	N	Ifosfamide	-	RT	43% ^‡^	85% ^‡^	62% ^‡^	49% ^‡^
Laack et al., (2011) [35]	223	Phase II	36	1.5	6	N	Cyclophosphamide, doxorubicin, vincristine, dexamethasone, carmustine, cytarabine	-	RT	56%	50%	38%	31%
Morris et al., (2013) [12]	135	Multicenter Phase II	52	3.5	5	Y	Procarbazine, vincristine	ICV ^a^	RT + HDCT (ARA-C)	95%	65%	57%	51%
O’Brien et al., (2000) [36]	136	Multicenter Phase II	46	1	1	N	-	IT ^e^	RT	95%	79%	65%	55%
Olivier et al., (2014) [37]	238	Multicenter Phase I	35	3	3	N	Idarubicin, vindesine, prednisolone	IT ^c^	RT	51%	55%	33%	28%
Omuro et al., (2015) [38]	105	Phase II	32	3.5	5 to 7	Y	Procarbazine, vincristine	-	SCT	94%	82%	79%	79%
Omuro et al., (2015) [39]	264A	Randomized Phase II	48	3.5	3	N	Temozolomide	-	HDCT (ARA-C)	71%	36%	28%	22%
Omuro et al., (2015) [39]	264B	Randomized Phase II	47	3.5	3	N	Procarbazine, vincristine, cytarabine	-	HDCT (ARA-C)	82%	36%	30%	20%
Pels et al., (2003) [40]	108	Pilot and Phase II	65	5	4	N	Vincristine, ifosfamide, dexamethasone, cyclophosphamide, cytarabine, vindesine	ICV ^a^	-	71%	-	-	-
Salamoon et al., (2013) [41]	249	Single-arm	40	3	6	N	Cytarabine, temozolamide	-	-	100%	100%	97%	95%
Shah et al., (2007) [18]	115	Single-arm	30	3.5	5 to 7 cycles	Y	Procarbazine, vincristine	IT ^a^	RT + HDCT (ARA-C)	92%	69%	57%	57%
Wieduwilt et al., (2012) [42]	165	Single-arm	31	8	Up to 8	Y	Temozolamide	-	HDCT(ARA-C, VP-16)	58%	53%	45%	45%
Total (N)	35		2115			Y = 11		14	29				

N^0^, number; HDMTX, high-dose methotrexate; Y, yes; N, no; CNS, central nervous system; ORR, overall response rate; CR, complete response; PR, partial response; PFS, progression-free survival; IT, intrathecal; ICV, intracerebroventricular; RT: radiation therapy; HDCT, high-dose chemotherapy; SCT, stem cell transplant; MPD, methylprednisolone; MTX, methotrexate; ARA-C, cytarabine; TMZ, temozolamide; VP-16, etoposide; **^†^** Unique cohort identification number; letter (A, B, C) following the number denotes a separate randomized cohort within a single trial. This number corresponds to the number in parentheses presented in the forest plots for Figure 2, Figure 3, Figure 4, Figure 5, Figure 6 and Figure 7; * Whole-brain radiotherapy or autologous stem-cell transplantation as consolidation strategies after high-dose methotrexate; ^#^ Survival data, time to recurrence (duration between date of diagnosis to date of death or last follow-up), was extracted from survival curves; ^¶^ Survival data, failure-free survival (duration between randomization date to relapse, progression, death, or last follow-up), was extracted from survival curves; ^ǁ^ 65 of 107 patients (60.7%) had a final scan after 3 cycles; 38 patients achieved CR, 15 achieved PR; ^‡^ Disease response for patients who achieved CR (no information available for those that achieved PR); PFS data extracted from survival curves for patients that achieved CR; ^a^ MTX; ^b^ MTX + ARA-C + MPD; ^c^ MTX + ARA-C + hydrocortisone; ^d^ MTX + ARA-C; ^e^ ARA-C (only if CSF positive).

**Table 2 cancers-15-01459-t002:** Pooled estimates for ORR and PFS among the cohorts categorized by number of planned courses, CNS chemotherapy, radiation therapy, and stem cell transplant.

		ORR (95% C.I.)	2-Year PFS (95% C.I.)
N^0^ of planned HDMTX courses (induction)	<5 (n = 20)	73% (65–80%)	50% (43–58%)
	≥5 (n = 15)	79% (66–89%)	52% (43–61%)
Chemotherapy to CNS compartment (induction)	No (n = 21)	73% (64–81%)	51% (42–59%)
	Yes (n = 14)	79% (70–81%)	52% (45–58%)
Radiation therapy (consolidation)	No (n = 13)	77% (65–87%)	52% (39–65%)
	Yes (N = 22)	74% (65–82%)	50% (45–56%)
Stem cell transplant (consolidation)	No (n = 29)	81% (68–92%)	59% (43–74%)
	Yes (n = 6)	74% (66–81%)	49% (43–56%)

ORR, overall response rate; PFS, progression-free survival; N^0^, number.

## Data Availability

The data presented in this study are openly available at: https://networks.resonancehealth.org/networks/hdmtx/media/hdmtx-pcnsl-systematic-review-data.

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
