# Peer review of "A Systematic Review of High-Dose Methotrexate for Adults with Primary Central Nervous System Lymphoma"

_cancers, 2023, doi:10.3390/cancers15051459_

Round 1

Reviewer 1 Report

The manuscript is an exhaustive review of the several schedules of administration of High Dose (HD) methotrexate (MTX) published for treatment of primary central nervous system lymphoma. The paper is well written and easy to read. The work is clinically relevant to determine whether there is a need to administer the highest dose of MTX. According to the results reported, it seems that intermediate doses are adequate. However, the authors should better address the following issues.

1.     1  Different doses of MTX have been used in the 26 articles identified. It would be interesting to know what the chronological trend of these uses is: continuous increase of MTX doses versus years, or, on the contrary, de-escalation, … or no trend at all?

2.    2   The authors should address this question: is MTX pharmacokinetics dose-dependent? If not, this means that average plasma MTX concentrations are proportional to the dose. If this is not the case, interpretation of the results reported in this paper (efficacy vs. dose) should be interpreted differently.

3.     3  Together with the MTX dose of each protocol, it would be interesting to know what the rescue protocols were: i.e., leucovorin dose according to post-infusion observed plasma MTX concentrations? In other words, can we imagine that the similar efficacy between low, intermediate, and higher MTX doses is due to higher leucovorin doses for the latter.

4.    4   Were patients over 70 years of age treated by lower MTX doses than younger ones? Indeed, a lower dose in elderly patients may be associated with similar plasma MTX exposure than that following a higher dose in younger patients, due to the reduced renal function of the former.

5.     5  Were intervals between cycles the same for every HD MTX protocols?

Other points:

1.     1  Why were articles written in Spanish or German (and not those in French, Russian, …) selected?

2.    2   What does “CNS chemotherapy” mean (bottom of page 6)? intrathechal or intracerebroventricular administration? Of which drug(s)? Only MTX?

Author Response

Responses to Reviewers

We very much appreciate the reviewers’ questions and comments regarding this systematic review. Based on the comments, we have reviewed our data and conducted additional analyses to address the reviewer’s comments and questions. Some of the new data have been incorporated into the main text or as supplemental tables; however, other data and explanations are presented only in this Response to Reviewers so as not to distract from the main message of this manuscript. We defer to the editorial board the decision to add the additional data to the main text.

Please see our responses below.

Reviewer 1

The manuscript is an exhaustive review of the several schedules of administration of High Dose (HD) methotrexate (MTX) published for treatment of primary central nervous system lymphoma. The paper is well written and easy to read. The work is clinically relevant to determine whether there is a need to administer the highest dose of MTX. According to the results reported, it seems that intermediate doses are adequate. However, the authors should better address the following issues.

  1. Different doses of MTX have been used in the 26 articles identified. It would be interesting to know what the chronological trend of these uses is: continuous increase of MTX doses versus years, or, on the contrary, de-escalation, … or no trend at all?

Thank you for this question. The publication dates for the articles included in this review span a period between 1992 and 2021, and over that time span we found no significant linear trend between MTX dose and chronological passage of time. We have added the following in the Discussion section:

“The publication dates for the articles included in this review span a period between 1992 (DeAngelis et al) and 2021 (Fu et al). Over the nearly 30 years that encompass this period there was no significant linear trend found between HDMTX dose and the chronological passage of time; most regimens used 3 – 3.5 g/m2 reflecting the fact that this dose has been and continues to be efficacious and safe in the treatment of PCNSL.”

  1. The authors should address this question: is MTX pharmacokinetics dose-dependent? If not, this means that average plasma MTX concentrations are proportional to the dose. If this is not the case, interpretation of the results reported in this paper (efficacy vs. dose) should be interpreted differently.

HDMTX was administered intravenously in all cohorts, and thus plasma methotrexate levels were proportional to dose. There can be, however, considerable inter-patient variability in clearance of HDMTX with severe delayed methotrexate clearance seen in 0.5 - 1% of pediatric patients with ALL (Taylor et al, 2020, https://doi.org/10.1002/cpt.1957). This elimination is predominantly affected by methotrexate-induced renal impairment that is seen in some patients. A study of real-world data from the Guardian Research Network (Ibarra et al, 2022, https://doi.org/10.1111/bcp.15506) suggested that a three compartment model, consisting of regions of the body that show similar kinetics, adequately describes methotrexate disposition. Although this is an important aspect of methotrexate efficacy and toxicity, we believe this is beyond the scope of this manuscript, and we have added the following to the Discussion to address the reviewer’s question:

“(As HDMTX was administered intravenously in all cohorts, peak plasma methotrexate levels were proportional to the HDMTX dose. This suggests that outcomes are closely related to the dose of methotrexate.) ”

  1. Together with the MTX dose of each protocol, it would be interesting to know what the rescue protocols were: i.e., leucovorin dose according to post-infusion observed plasma MTX concentrations? In other words, can we imagine that the similar efficacy between low, intermediate, and higher MTX doses is due to higher leucovorin doses for the latter.

We agree that leucovorin rescue is important for HDMTX-backbone regimens, and most of the articles reviewed did describe administration of leucovorin as a component of supportive care. However, not all articles provided specific details regarding leucovorin dose, timing, or duration, and there was considerable variation in leucovorin administration protocols among those that did provide that information. We have added the a supplementary table (Table S2) describing leucovorin rescue doses for the study cohorts and the following text in the Limitations portion of the DISCUSSION:

“Additionally, there was considerable variability among the treatment cohorts with respect to the administration of leucovorin rescue (specific dose, timing, duration) which would have potentially impacted the frequency of toxicities as well as treatment efficacy (Supplementary Table S2).”

  1. Were patients over 70 years of age treated by lower MTX doses than younger ones? Indeed, a lower dose in elderly patients may be associated with similar plasma MTX exposure than that following a higher dose in younger patients, due to the reduced renal function of the former.

We agree there might have been variations in HDMTX dose for elderly patients to accommodate their reduced physiologic function. Patients 70 years and over were included in 15 of the 35 cohorts; however, due to the lack of patient-level data (i.e., only median age and age range were reported for each cohort, and the number of patients over 70 years within each cohort was not presented) we were unable to assess the relationship between HDMTX dose and age in this review. These 15 cohorts received the same proportion of low (<3 g/m2), intermediate (3-4.9 g/m2), and high (≥5 g/m2) doses as the 20 cohorts that did not include patients over 70 years of age. While many patients may have received dose reduction due to previous toxicities, we could not determine whether there were any a priori dose reductions based on patient age. As we have mentioned in the DISCUSSION section, data on elderly patients with PCNSL is lacking and we believe more work should be done investigating treatment in this population. 

  1. Were intervals between cycles the same for every HD MTX protocols?

No, there was much variability among the HDMTX protocols reported for each cohort and the intervals between cycles were also variable. The time between doses of MTX ranged from 1- 3 weeks.

In cases where it was specified, 14 days was the most commonly planned time between courses (64.7%), followed by 7 (17.6%) and 10 (11.8%) days. Planned dose per cycle showed a linear relationship with the planned dose density (total amount of planned methotrexate / total number of planned days of treatment) with higher dose regimes having a higher dose density. Dose is a proxy of the intensity of treatment. Planned dose was proportional to the total amount of MTX given as indicated by the graph below.

Other points:

Why were articles written in Spanish or German (and not those in French, Russian, …) selected?

Thank you for this question. For clarification, our initial search included all languages and after reviewing for inclusion/ exclusion criteria, the remaining manuscripts were all either English, Spanish or German. This has been clarified in the text as follows:

“Criteria for inclusion were clinical studies on CNS tumors in humans and use of HDMTX (defined as dose ≥ 500 mg/m2). Criteria for exclusion were non-clinical or animal studies and review articles. The initial search identified 587 articles pertaining to the use of HDMTX in CNS malignancies, all of which were written in English, Spanish or German. Following removal of duplications, four authors (MG, GV, CS, PS) independently assessed…”

What does “CNS chemotherapy” mean (bottom of page 6)? intrathecal or intracerebroventricular administration? Of which drug(s)? Only MTX?

CNS chemotherapy refers to “chemotherapy to the CNS compartment (i.e., via intrathecal or intracerebroventricular administration) during induction” as explained in the Data Collection portion of the METHODS section. A total of 14 cohorts received CNS chemotherapy (Table 1). Specific drugs delivered were methotrexate (MTX), cytarabine (ARA-C), methylprednisolone (MPD), or hydrocortisone.

We have added superscripts to the “IT or ICV” in the “CNS delivery” column for the specific drugs denoted in the footnotes for Table 1 to provide detailed information on CNS chemotherapy.

Reviewer 2 Report

Review papers on therapy and clinical trials are valuable scientific material. The manuscript "A systematic review of high-dose methotrexate for adults with primary central nervous system lymphoma" presented to me for review is properly prepared and transparent. I am pleased to recommend the article for further editorial processing.

Author Response

The authors appreciate this positive review of our manuscript.

Reviewer 3 Report

The systematic review of Inés-Villanueva et al. presents the analysis of the current therapeutic approaches with high-dose methotrexate for primary central nervous system lymphoma in adults. That subject is always up-to-date, because of the malignancy of the disease and the only sufficiently effective treatment (with methotrexate).

The analysis was done according to PRISMA guidelines, and the results clearly presented.

The results support high-dose treatment, especially with the addition of co-drugs, but also show the areas that should be more elaborated (older patients).

I would just suggest reviewing the ms to correct some vocabulary/grammar/spelling/punctuation mistakes.

Author Response

Thank you for these comments and for reviewing the document. The authorship consists of native American and British English speakers. We chose American English spelling and punctuation throughout for consistency. We have reviewed the document again to identify and correct any potential errors. The only potential source of debate might be the use of the ‘Oxford comma’ in this document, but the authors believe this is a matter of personal preference rather than incorrect vocabulary, grammar, spelling, or punctuation.

Round 2

Reviewer 1 Report

Thank you for the additional information you have included in the revised version.